# Transient Secondary Structures as General Target-Binding Motifs in Intrinsically Disordered Proteins

**DOI:** 10.3390/ijms19113614

**Published:** 2018-11-15

**Authors:** Do-Hyoung Kim, Kyou-Hoon Han

**Affiliations:** Genome Editing Research Center, Korea Research Institute of Bioscience and Biotechnology, 125, Gwahak-ro, Yuseong-gu, Daejeon 34141, Korea; organic2@kribb.re.kr

**Keywords:** intrinsically disordered protein (IDP), pre-populated, transient secondary structure, pre-structured

## Abstract

Intrinsically disordered proteins (IDPs) are unorthodox proteins that do not form three-dimensional structures under non-denaturing conditions, but perform important biological functions. In addition, IDPs are associated with many critical diseases including cancers, neurodegenerative diseases, and viral diseases. Due to the generic name of “unstructured” proteins used for IDPs in the early days, the notion that IDPs would be completely unstructured down to the level of secondary structures has prevailed for a long time. During the last two decades, ample evidence has been accumulated showing that IDPs in their target-free state are pre-populated with transient secondary structures critical for target binding. Nevertheless, such a message did not seem to have reached with sufficient clarity to the IDP or protein science community largely because similar but different expressions were used to denote the fundamentally same phenomenon of presence of such transient secondary structures, which is not surprising for a quickly evolving field. Here, we summarize the critical roles that these transient secondary structures play for diverse functions of IDPs by describing how various expressions referring to transient secondary structures have been used in different contexts.

## 1. Introduction

Intrinsically unstructured/unfolded proteins (IUPs), now known more as intrinsically disordered proteins (IDPs), exist in a structural state that resembles a pre-molten globule or a random coil, i.e., a state where no 3-D structure is present [1]. The disorder in IDPs is different from the long-known protein disorder which refers to short flexible linkers or loops in globular proteins that are typically composed of less than 20 amino acids. The DisProt DB holds ~ 800 proteins, where IDPs as well as many proteins with short disordered fragments are listed [2]. An IDP or an intrinsically disordered region (IDR) in an IDP consists of at least 40 amino acids, often exceeding 100 residues [3]. Clearly, it is interesting that such a long stretch of amino acid residues can be “unstructured”, unable to fold into a tertiary structure, under pseudo-physiological conditions. A more novel and surprising fact is that IDPs/IDRs can perform diverse biological functions under such a structural state, which has drawn a lot of attention along with another fascinating fact that these proteins or their IDRs are closely associated with fatal diseases such as cancers, Alzheimer disease (AD), Parkinson’s disease (PD), and viral pandemics [4,5,6,7,8]. Many transcription factors (TF) are hybrid-type IDPs where intrinsically disordered transactivation domains (TADs) coexist with globularly structured DNA-binding domains [9,10,11]. For example, as high as 49% among 401 human TFs in the Swiss-Prot database contains long IDRs [12]. The well-known causative agents for AD and PD are tau and alpha-synuclein, respectively, and both are IDPs.

For decades, textbook knowledge for protein–protein interactions have been based on the lock and key paradigm. However, discovery of a large number (thousands) of IDPs/IDRs has utterly destroyed such a classical paradigm, 3-D structure = function, in protein science and structural biology and has deeply reshaped our view on proteins by showing that IDPs can promiscuously bind to their targets without 3-D structures or without the active pockets afforded by such 3-D structures. A massive amount of bioinformatics prediction studies suggest that as much as 50% of entire human proteome may be IDPs or contain IDRs [3]. A remarkable fact that the fraction of IDPs is higher in eukaryotes than in lower organisms even suggests that IDPs may be a more evolved form of proteins. In a sense, IDPs are not different from globular proteins in that they need to bind their targets in order to manifest their biological signals. Yet, only IDPs can promiscuously interact with many partners, often serving as hubs in protein–protein interaction networks (PPIN) [13,14]. Hence, our knowledge on PPIN gained without considering IDPs would be quite incomplete and at best half a piece.

Even though several subtypes of IDPs with different functions exist [15], a common denominator among all subtypes is that they must be able to bind their targets which may be proteins, nucleic acids, lipids and metals. Thus, these novel and unorthodox IDPs are leading us to confront and answer a fundamental question of how they bind to targets without an active pocket. Are there any “specificity determinants” within IDPs that mediate and control their binding to targets? No satisfactory answer has been obtained to this question even after two decades of rather intense research. One school of thought has been arguing from the early days that the overall disordered nature itself is important for function [16]. This school advocated an idea that a secondary structure, e.g., a helix, should be induced only upon binding of IDPs to targets [17]. For example, in the nuclear magnetic resonance (NMR) investigation of the kinase inducible domain (KID) of CREB that binds with KIX domain of the coactivator CREB binding protein (CBP) the authors described that the two amphipathic alpha helices of KID seen in the KID–KIX complex were formed only after binding of KID with KIX [16], but Hua et al., [18] and Radhakrishnan et al. [19] showed one year later that these two helices are ~10% and ~50% pre-populated prior to KIX binding. The expressions such as a disorder-to-order transition [20] and binding coupled folding [21] have also been in use along this line of thought, but these expressions are rather qualitative as they tend to depict only the overall topological change of IDPs upon target binding. This school was heavily inclined to emphasize an induced fit (IF) mechanism as the only plausible mechanism for IDP-target binding based on the conclusions that IDPs were completely unstructured (CU) [16,22].

On the other hand, another school of thought proposed that IDPs would contain some sort of specificity determinants such as transient secondary structures and these structures pre-exist or pre-form within free IDPs prior to target binding and serve as target recognition antennae. This school opened a door to another possible mechanism, i.e., conformational selection (CS) for IDP-target binding. Even though a few pioneering NMR reports addressed this point [23,24,25], this second school was not acknowledged [26] as vividly as it should have been until recently. In order to provide an overview on transient secondary structures in IDPs/IDRs, the significance of which the second school of thought has been addressing over the last 20 years, we summarize here how functionally important transient structural features in free IDPs/IDRs have been described. We acknowledge that our literature survey may not be complete, but it is sufficient to deliver a feel for the trend.

## 2. Results

### 2.1. Favoring or Preferring Helical Conformation, Nascent Helix, and Helix-Forming

The Anti-sigma transcriptional factor FlgM is an inhibitor of a transcription factor σ28 that is specific for the expression of bacterial flagella and chemotaxis genes. Daughdrill et al. reported that the C-terminal half of FlgM is disordered, but exists in a non-random structural state that was described as mostly unfolded state or a partially collapsed state [27]. This 57-residue C-terminal disordered region was shown to have two sequential regions M60–G73 and A83–A90 favoring helical conformations. The chemical shifts of these residues deviated significantly from the random coil values. These helix-favoring residues were known to be essential for the anti-σ28 activity of FlgM.

Hua et al. reported that an unbound phosphorylated KID (pKID) contained two nascent helical segments presence of which was evidenced by small but significant deviations of ^1^H NMR chemical shifts at 4 °C from random-coil values [18]. In this report the authors even included a section in Results called “Intrinsic Helical Propensities” where they explained in detail how the interproton NOEs also support their conclusion (see in [18]). Most critically, they were able to detect a contiguous series of dαN(i,i + 3) and dαβ(i,i + 3) NOEs for the αA helix (residues 119–130) in aqueous solution both for phosphorylated and non-phosphorylated KID. The two nascent helical segments, αA helix (residues 119–130) and αB helix (residues 134–141), were shown to correspond to the well-defined recognition helices observed in the structure of the pKID-CBP complex [16]. This work drastically contrasted to an earlier report which described the unbound KID to be in a random coil state [16].

Rudolph et al. reported that the Cdc42/Rac binding region of the Wiskott Aldrich syndrome protein (WASP) has a 13-residue helix-forming segment (residues 252–264) [28]. A homonuclear 2D NMR study on the anchor domain of Nef of HIV-1 by Geyer et al. showed that the domain is not folded but has a short two-turn-helix (R35–G41) and another potential helix (P14–R22) [29]. Alpha-synuclein (α-Syn) is an important IDP associated with Parkinson’s disease and is largely unfolded in solution. Eliezer et al. demonstrated that α-Syn has a region with a preference for helical conformation which could be important in the aggregation of α-Syn into fibrils [25]. They described this preference for helical structure as a nascent or transient helix. The term nascent helix was also used by Tharpar et al. who found that the 91-residue N-terminal IDR (G17–K108) of dSLBP (stem-loop binding protein from Drosophila) contain four regions (I28–A45, S50–L57, S66–G75, and F91–N96) with a helical propensity [30]. They used other NMR parameters, such as heteronuclear NOEs, amide–amide interproton NOEs, and protection from solvent exchange in addition to chemical shifts. The four regions of this IDR that form nascent helices had small but positive heteronuclear NOEs, and sequential d_NN_ interproton NOEs even though medium- and long-range interproton NOEs were absent. In addition, small but significant (larger than ten-fold) protection from solvent exchange was observed for the helical regions. 

Jensen et al. reported that the partially folded C-terminal domain of Sendai virus nucleoprotein (N_TAIL_) preferentially populates three specific overlapping helical conformers [31]. They have measured residual dipolar coupling constants (RDC) and performed Flexible–Mecanno calculation to come up with location of three overlapping helices. Gupta et al. reported that the intrinsically unstructured domain 3 (residues 359–447) of NS5A in Hepatitis C virus of appears to have helical conformations weakly populated over two segments: S401-S412 and D427-V445 [32].

### 2.2. Local Structural Elements

Lee et al. reported that the 73-residue transcriptional activation domain (TAD) of an important tumor suppressor (p53) is not folded, i.e., mostly unstructured, and yet contains local structural elements such as a transient amphipathic helix and two nascent turns [23]. They noticed that these local secondary structures are formed by functionally and positionally critical hydrophobic residues and proposed that such structural elements should represent specificity determinants important for the transcriptional activity of acidic TADs. Among the three local structural elements the pre-populated (~30%) amphipathic helix in the unbound p53 TAD exactly overlapped with the helix found in the complex structure of a p53 peptide (residues 15–29) with the N-terminal domain (residues 3–109) of mdm2 [23]. This work contributed significantly to ending a long controversy in the transcription field as to whether a transactivation domain should contain a helix at the TAD-target binding interface. The authors underlined the argument that the structural transition associated with binding of the p53 TAD to mdm2 would not be a coil-to-helix type as was claimed in the KID–KIX binding [16], but should just involve tightening of the preexisting helix into a stable helix as the former helix is almost identical, formed by exactly the same residues, to the helix observed in the mdm2-bound state.

Sml1 is a small protein (104 residues) that binds to the large subunit of ribonucleotide reductase (Rnr1). This interaction between Sml1 and RNR not only inhibits RNR activity, but also results in lethality to *mec1* or *rad53* that are essential for yeast cell growth. Zhao et al., found that this lethality is suppressed by removal of Sml1 and also showed using NMR that Sml1 was highly flexible in solution and had a loosely folded tertiary structure [24]. They also found that Sml1 contains three local structural elements including a long alpha-helix (residues 61–80) within its 33-residue C-terminal region. Mutation of four residues that belong to this helix critically affected the Sml1 activity; all four mutations abolish the Sml1 interaction with Rnr1. It is interesting that two independent reports, one on the p53TAD and the other on Sml1, used the same term of local structural elements to denote transient secondary structures in unbound IDPs.

ExsE is a small 81-residue protein that controls T3SS gene expression in *Pseudomonas aeruginosa*. Zheng et al., found that ExsE is largely unfolded and such conformation of ExsE may expedite efficient secretion through the narrow path of the T3SS secretion channel [33]. The title of this report is “the transiently ordered regions in intrinsically disordered ExsE are correlated with structural elements involved in chaperone binding”. The authors concluded that preexisting structured elements facilitate binding of IDPs to their targets based on their observation that ExsE contains localized structurally ordered regions which are related to the secondary structures observed in the crystal structure of the ExsE-ExsC complex. Lum et al., studied segmental chain motions in p53TAD using fluorescence quenching and fluorescence correlation spectroscopy (PET-FCS) [34]. In this report the authors used wordings like dynamic local structure, distinct elements of secondary structure and local helical structural element. NMR characterization on the C-terminus of ICln (residues Q159–H235) by Schedlbauer et al. revealed that this IDR adopts a preferred α-helical organization between residues E170 and E187, and the residues D161–Y168 and E217–T223 exist preferentially in extended conformations (β-strands) [35]. The title of this report carries a wording of “local structural preformation”.

### 2.3. Preformed, Pre-Ordered, Pre-Organized, and Pre-Structured

The C-terminal part (residues 25–96) in the 96-residue cytoplasmic region of synaptobrevin-2 is important for formation of SNARE complex. This 96-residue IDR was described to be largely unfolded in solution, but had a preformed nascent helix (residues 78–91) [36]. Intuitively, one can understand the structural resemblance between the mostly or largely unfolded state of IDPs and the folding initiation states populated with transient secondary structures. In the folding study of a 33-residue homodimeric coiled-coil peptide GCN4-p1 Zitzewitz et al. found that helix-helix interactions between the preformed transient helices were important for the folding of the GCN4-p1 [37]. The 47-residue intrinsically disordered cytoplasmic tail of the amyloid precursor protein (APP) is important for intracellular signaling and proteolytic processing of APP. Ramelot et al. used the key words such as transient and preordering in the title of their report to describe their findings [38]. Ribosomal protein S4 (RPS4) is one of the first proteins to interact with rRNA in the process of ribosome assembly. Its N-terminal 41 residues is known to be necessary both for the assembly of functional ribosomes and for full binding to 16S RNA. While the N-terminal is highly flexible in solution without discernible secondary structure Sayers et al. [39] observed two short segments S12-L15 and P30-P33 in RPS4 forming transiently ordered states and published their results with a title including structural preordering. A structural study on p27^Kip1^ by Bienkiewicz et al. was published with a title of “functional consequences of preorganized helical structure in the intrinsically disordered cell-cycle inhibitor” [40]. This investigation showed that p27^Kip1^ exists in a largely unfolded state and contains marginally stable helical structure that presages the R-helix formed when p27^Kip1^ binds to cyclin A-Cdk2.

The preS1 surface antigen of hepatitis B virus is associated with hepatocyte binding and hence important for viral entry. Chi et al. published an article with a title of “Pre-structured motifs in the natively unstructured preS1 surface antigen of hepatitis B virus” where they investigated the structural features of the 119-residue full-length preS1 using heteronuclear NMR methods and discovered that its N-terminal 50 residues are populated with multiple pre-structured motifs. The most prominent motif formed by residues 27–45 overlaps with the putative hepatocyte-binding domain [41]. The term pre-structured motif was used in later studies to describe transient second structures. For example, using N-15 heteronuclear NMR experiments and replica exchange molecular dynamics simulations Lee et al., investigated the structural features of the flexible N-terminal region (46 residues) of E7, the early oncogene product of the high-risk human papillomavirus (HPV) 16 in order to understand how E7 contributes to the transforming activity. Several NMR parameters such as chemical shift indices (CSI) or secondary structure propensity (SSP) scores, ^15^N-^1^H heteronuclear NOEs, T_1_ and T_2_ relaxation times, and temperature coefficients of backbone amide protons were measured. The ensemble structures obtained by replica exchange molecular dynamics (REMD) simulation which incorporated NMR parameters indicated that this IDR of E7 contains two transient (10–20% populated) helical pre-structured motifs that overlap with important target binding moieties such as an E2F-mimic motif and a pRb-binding LXCXE segment [42].

p53 is an important tumor-suppressor protein, the deactivation of which by mdm2 results in cancers. A SUMO specific protease 4 (SUSP4) was shown to rescue p53 from mdm2-mediated deactivation [43]. In order to understand the p53 rescue mechanism by SUSP4 Kim et al. studied the structure of a mid-domain (residues 210–300) of SUSP4 that is responsible for p53 rescue [44]. The authors found that this 100-residue IDR contains a 29-residue pre-structured motif which they named “p53 rescue motif” since this motif binds to mdm2 and disrupts p53-mdm2 binding and exhibits an antitumor activity in cancer cell lines expressing wild-type p53. This p53 rescue motif contains two transient helices connected by a hydrophobic linker. A recent NMR study on the intrinsically disordered tau1core transactivation domain of human glucocorticoid receptor showed that this intrinsically disordered activation domain contains three helix pre-structured motifs with ~30% pre-population [45]. The eIF4E-binding protein (4EBP1) has been a symbol of a completely unstructured type IDP for two decades because it was mistakenly shown not to contain any local order due to an incomplete resonance assignment in 1998 [22]. A recent NMR study revoked this belief by showing that its eIF4E-binding fragment (residues 55–63, i.e., “active” residues) forms a pre-structured helix [46].

### 2.4. Transient Structure and Transient Order

By combining results from NMR, CD and SAXS Moncoq et al. showed that phosphorylated insulin receptor interaction region (PIR) lacks ordered structure, but has a short stretch (residues 399–407) which is structured transiently [47]. The oncoprotein c-Myc is a key regulator of cell growth and is an important target for drug development. Andersen et al. carried out an NMR investigation on an N-terminal IDR (residues 1–88) of c-Myc and found multiple regions including two helices that are transiently structured [48]. In an effort to better understand of the role of the NS5A protein during hepatitis C virus infection Feuerstein et al., used resolution-optimized multidimensional NMR methods complemented by small-angle X-ray scattering methods and obtained detailed atomic resolution information on transient local and long-range structure of intrinsically disordered domain 3 (residues 191–369) of NS5A in Hepatitis C virus [49]. They learned that, despite being largely disordered, this 188-residue IDR contains three regions that transiently adopt α-helical structures, partly stabilized by long-range tertiary interactions. Two of these transient α-helices form a noncanonical SH3-binding motif, which allows low-affinity SH3 binding. They also characterized two distinct interaction modes with the SH3 domain of Bin1 (bridging integrator protein 1), a pro-apoptotic tumor suppressor. This report was published with a title of “Transient structure and SH3 interaction sites in an intrinsically disordered fragment of the Hepatitis C virus protein NS5A”. The term transiently ordered regions was also used by Zheng et al. [32].

### 2.5. Other Expressions

Laptenko and Prives described the transient secondary structures found in p53TAD as small islands of secondary structures [50]. Mittag et al., used a term minimal ordering of short linear motifs [51]. Kim et al., used a term residual secondary structural elements to describe hTAF(II)31-binding helical motif in the intrinsically unfolded transcriptional activation domain of VP16 [52]. Novacek et al. used a slightly different term of a transient secondary structure in their NMR study on the microtubule associated protein 2c [53]. 

### 2.6. An overall Topology of IDPs

In addition to above expressions that describe local structural elements in IDPs/IDRs several terms have been coined in order to describe the overall topological state of IDPs such as mostly unfolded, partially collapsed, largely unfolded, mostly unstructured (MU), and loosely folded. All these terms are in contrast to the term completely unstructured (CU).

## 3. Discussion

We want to mention that the above sections in Results were divided for convenience as many reports used more than one type of expression. For example, the report on p53TAD not only used the terms local structural elements and mostly unstructured (MU), but also pre-existing minimal secondary structures, loosely folded, largely unstructured, local structural order, etc. [23]. The bottom-line message should be that many IDPs/IDRs exist in the mostly/largely unfolded, mostly unstructured, or loosely unfolded state that are pre-populated with transient secondary structures. A recent analysis shows that as much as 70% of all the IDPs characterized by NMR contain these transient structures [6]. Judging from the fact that some IDPs such as VP16 TAD, 4EBP1, 4EBP2 and securin originally reported to be in a CU state before the PreSMo concept was introduced turned out to be in a MU state by later studies [52,53,54,55,56] there is a good possibility that some of the CU type IDPs studied in the early days may turn out to be in a MU state. It is unlikely that certain IDPs composed of simple dipeptide repeats involved in phase separation or associated with formation of membrane-less organelles would contain such transient structures. Nevertheless, it is noteworthy that even simple repeat IDPs such as polyglutamine and polyproline are able to form an α-helix [57] and a PPII helix [7], respectively. Interestingly, transient secondary structures in IDPs/IDRs are often observed in tandem [5,23], suggesting that some sort of intramolecular long-range synergistic interactions among multiple motifs is possible as detected by single-molecule FRET or PRE experiments [23,49,58].

Since a large number of IDPs/IDRs contain transient secondary structures as essential features for target binding and more of such structures are expected to be discovered in the future, we have recently proposed to designate them with a more fitting name of “Pre-Structured Motifs (PreSMos) [5,6]. This term is more explicit than bioinformatics terms such as molecular recognition element (MoRE) or pre-formed structural element (PSE) [59,60] in that it reveals if a particular segment within IDPs/IDRs is pre-structured or not. It appears that a PreSMo in certain IDPs may even serve as the initial aggregation hot spot [25,61]. NMR is highly robust and unique as described below in that the precise location in terms of amino acid numbers as well as the degree of pre-population of PreSMos in percentile values can be attained only by NMR measurements [5,6]. NMR chemical shift indices (CSI) or secondary structure propensity (SSP) scores are usually sufficient to provide a quick judgmental criterion on the existence of a PreSMo, but other NMR parameters such as interproton NOEs, heteronuclear NOEs, ^3^J coupling constants, relaxation times, temperature coefficients of backbone amide protons, and hydrogen exchange rates, can be used additionally, if available, to ratify the conclusion drawn by chemical shifts and to better understand the transient and dynamic nature of a PreSMo. Table 1 summarizes NMR parameters measured to delineate transient secondary structures in several MU type IDPs. Even though CSI and SSP values are available for all IDPs other common parameters such as interproton NOEs, heteronuclear NOEs, T_1_ and T_2_/T_1ρ_ relaxation times, are available for ~ 1/2 of the IDPs studied. The ^3^J_HNHα_ coupling constants that would indicate presence of secondary structures when they are less than 6 Hz or larger than 8 Hz are available only for 6 cases. The temperature coefficients of backbone amide protons which is associated with formation of a hydrogen bond in secondary structures when <5 ppb/deg are also available for seven cases. The hydrogen exchange rates are rarely reported since IDPs are not likely to have slow and hence measurable exchange rates due to lack of globular structures that could retard exchange process.

It seems worth mentioning that using the term “residual” for transient structures could be misleading since it implies that the MU state of IDPs were derived from a fully stable 100% secondary structural state. We should add that while the conclusion on the location and degree of pre-population of PreSMos in all IDPs/IDRs described above is conclusive, there is one exception of α-Syn. No consensus has been reached yet regarding the structural nature of this important IDP even though the verdict is leaning towards the presence of pre-structuring [25,62,63]. In a few cases the structural details of both a PreSMo and its target-bound conformation are available, which illustrate how remarkably PreSMos presage target-bound conformations. Well-known examples are p53TAD/mdm2 [23,64], KID/KIX [16,18], and 4EBP1/4EBP2 with eIF4E [53,56,65]. The concept of “fuzziness” was introduced for IDP-target complexes [66], which refers to a bioinformatics argument that the target-bound structure of an IDP fragment may not have fixed spatial and temporal coordinates. Apparently, this concept does not apply to the above well-known IDP-target complexes, since all the target-bound PreSMo fragments in these examples have a unique helical conformation. The dynamic conformational nature (proposed as “dynamic” fuzziness) of the target-bound IDP fragment may be a relevant concept to the completely unstructured part that is outside of a PreSMo but is not applicable to the PreSMo part that actually makes contacts with target proteins through various interactions such as electrostatic, H-bonding, and hydrophobic interfaces, and becomes a MoRE or PSE [59,60]. It is implicitly included in the definition of a PreSMo that two flanking regions of any PreSMo in an IDP/IDR should be completely devoid of any secondary structures. There is a lingering question regarding their functional significance of PreSMos. In recent mutation studies on ACTR and p53TAD, this aspect was unambiguously verified where the degree of the pre-population of a PreSMo was shown to be critical for target binding [67,68]. In addition, either gain-of-function or loss-of-function mutants of hGR tau1core activation domain were all associated with mutations of PreSMo-forming residues, demonstrating the functional significance of PreSMos [69].

## 4. Conclusion and Perspective

Delineation of transient secondary structures or pre-structured motifs in target-unbound IDPs/IDRs has unequivocally shown that it, in fact, is the PreSMos, minimally looking transient secondary structures, that mediate target binding of IDPs, not the disordered nature per se of IDPs; even though IDPs do not form spatial active pockets, they do contain such specificity determinants that govern target binding [5,6]. It is quite unfortunate that the early day hasty conclusion drawn from only few reports has prevailed the IDP field for a long time with a strongly biased misconception that pre-structured motifs would not be present in IDPs or not needed for IDP function [26]. The fact that the pre-populated degree of PreSMos is subtly tied to target binding behavior of IDPs [67,68] enlightens us how transient and dynamic features are utilized, not just rigid 3-D shapes, of proteins in order to subtly control protein interaction signals. IDPs are not total outliers in protein structural biology as one may simply think, since IDPs comply well with the shape complementarity rule of binding established with globular proteins as one can observe the surfaces of PreSMos fitting into the binding grooves or pockets of globular target proteins [5].

It needs to be underlined that discovery of transient secondary structures or PreSMos in the unbound IDPs/IDRs critically contributed to balancing the IDP field in terms of target binding mechanisms. When free IDPs/IDRs were viewed to be completely unstructured, IF was the only option for the IDP-target binding mechanism. As PreSMos are appreciated more and more, many have become aware of the possibility that conformational selection would play a role in target binding and support a concerted mechanism for target binding of IDPs. Initial recognition of IDPs by target proteins may involve conformational selection of a PreSMo, whereas the late stage of binding would involve structural tightening of a PreSMo into a stable target-bound conformation as shown in Figure 1 [5,6,70].

Studies on IDPs will provide missing links in PPIN and consequent identification of potential markers for drug design either by blocking IDPs themselves, or by discovering inhibitors against the target proteins IDPs bind to. Small molecule inhibitors of IDPs were discovered recently [71,72]. Also, application of peptides derived from the pre-structured motifs have been shown to have a potential as anti-cancer peptide pharmaceuticals [44].

## Figures and Tables

**Figure 1 ijms-19-03614-f001:**
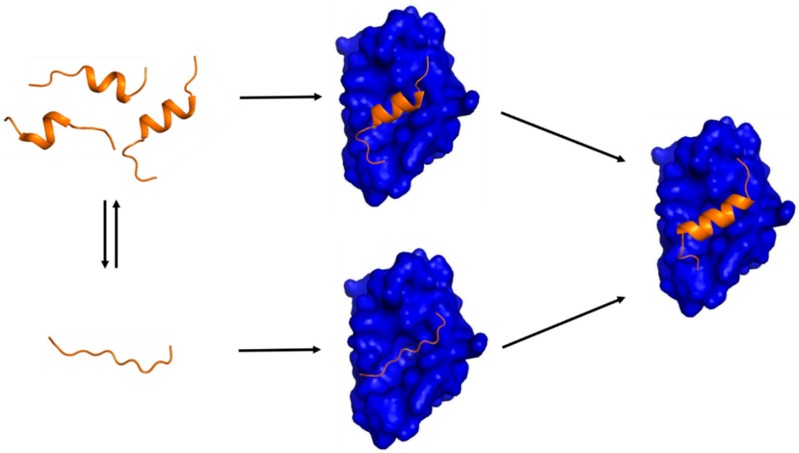
A schematic diagram showing two possible mechanisms of IDP-target binding. (**Left**) a target-free fragment of an IDP or a long (> 40 residues) IDR exists in a conformational ensemble where a completely unstructured state (bottom) is in equilibrium with a pre-populated state having transient helices of slightly different helical lengths (top). The fraction of the pre-populated state has been shown to vary from one IDP to another (10 ~ 70%) [5]. The three pre-structured helices with slightly different helical lengths were experimentally observed by Flexible-Meccano calculations using residual dipolar couplings in the case of the N_TAIL_ of Sendai virus nucleoprotein [31] and 4EBP1 [46]. (**Right**) a transient pre-structured helix seen in the target-unbound state becomes a stable helix upon target binding, which was observed in the case of p53TAD/mdm2 [23,64], KID/KIX [16,18], and 4EBP1/4EBP2 with eIF4E [46,56,65]. A peptide with a sequence of the pre-structured helix in p53TAD or in 4EBP1 is an inhibitor of mdm2 or eIF4E, which provides a basis for anti-cancer peptide therapeutics design. (**Middle**) two potential encounter complexes (EC). The top EC would be formed by a conformational selection (CS) process of a PreSMo by a target while the bottom EC produced in an induced fit (IF) mechanism. The top CS path would be more favorable in terms of entropy than the bottom path since the bottom EC would have to face a larger entropic penalty to form a stable helix during or upon target binding.

**Table 1 ijms-19-03614-t001:** NMR parameters used for delineating transient secondary structures or pre-structured motifs in mostly unstructured IDPs/IDRs.

	CSI/SSP	Interproton NOE	^15^N-^1^ HHet NOE	T_1_	T_2_ T_1ρ_	^3^J_HNHα_	Temp. Coeff. Backbone NH	HX Rate	Ref
FlgM	O		O	O	O				27
KID	O	O							18
GBD/CRIB WASP W7	O	O						O	28
HIV-1 Nef	O	O				O	O		29
Synaptobrevin-2	O								36
APPC	O	O				O			38
p53 TAD	O	O	O	O	O	O	O	O	23
RPS4	O		O	O	O				39
α-Synuclein	O								25
Securin	O		O	O	O				55
VP16 TAD	O	O							54
VP16 TAD	O	O	O	O	O	O			52
preS1 of HBV	O	O	O	O	O	O	O		41
Sml1	O								24
dSLBP	O		O	O	O				30
N_TAIL_ Sendai V.	O								31
nucleoprotein									
Sic1	O		O		O				51
c-Myc	O	O	O	O	O				48
ExsE	O		O	O	O				33
MAP2c	O								53
NS5A HCV	O	O							32
NS5A HCV	O		O	O	O				49
4EBP2	O		O		O				56
4EBP1	O	O	O	O	O		O		46
ICIn	O								35
TAU	O				O	O			7
E7 HPV	O		O	O	O		O		42
SUSP4	O	O	O	O	O		O		44
hGR tau1c	O	O	O	O	O		O		45
Huntingtin Httex1 25Q	O		O						57

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
