# Peer review of "Transient Secondary Structures as General Target-Binding Motifs in Intrinsically Disordered Proteins"

_ijms, 2018, doi:10.3390/ijms19113614_

Round 1
Reviewer 1 Report
The paper presents the screening of proteins with intrinsically disordered fragments - regions or ven complete proteins. The classification accoridng to the origin, functional meaning and folding process related events is interesting.
Major commenrs
However the paper - according to my idea - represents the form of meta-dta-analysis. I do not see any information about it. It should be given in the title of the paper or in the list of keywords
Minor comments:
Some abbreviations used lack explanation.
For example KIX - line 70
All headers of table 1 shuld be explained - for example in the legend of this Table.
Author Response
The paper presents the screening of proteins with intrinsically disordered fragments - regions or ven complete proteins. The classification accoridng to the origin, functional meaning and folding process related events is interesting.
Thank you for helpful comments.
Major commenrs
However the paper - according to my idea - represents the form of meta-dtaanalysis. I do not see any information about it. It should be given in the title of the paper or in the list of keywords
=> We have modified the title to reflect the comment.
Minor comments:
Some abbreviations used lack explanation.
For example KIX - line 70
=> Explanation is given.
All headers of table 1 shuld be explained - for example in the legend of this Table.
=> The parameters are the standard parameters used in NMR technology and hence we do not think that they need be explained.
Reviewer 2 Report
In this paper named “Transient secondary structures in intrinsically disordered proteins”, authors bring a very interesting topic of the presence of transient secondary structures in IDPs and they addressed several works of literature in this regards. They also mentioned precisely the importance of these pre-populated transient secondary structures in the fulfillment of the binding, recognition and ultimately the function. There are a few issues that I think to discuss more elaborately:
1) The authors put emphasis on pre-existing transient structure in free IDPs prior to target binding and they mentioned it as recognition antennae. I am wondering what it would be helpful for the reader to understand their point if they provide a well-defined mechanism for their proposal.
2) In this context, I would be interested to know about the author’s interpretation of the fuzzy complex of IDPs.
3) In Line 122, authors mentioned the David Eliezer’s paper to address the term “preference for helical structure “in alpha-synuclein. I am curious to know whether “preference for helical structure “is equivalent to “having stable helical secondary structure” (as in Eliezer et al paper, far UV spectrum of alpha-synuclein in absence of SDS showed random coil-like structure) and author’s explanation would be more useful for the general reader to understand this fine difference or similarities.
4) Authors demonstrated the “transient secondary structure in IDPs “in various literature very efficiently. In this literature, different scientists interpret their (preformed structures) existence according to their problem. If I try to generalize the importance of their presence in protein-protein interaction, then the knowledge is still elusive. In my opinion, more literature, if any, dealing with the acceleration of binding or enhance the specificity to the binding partner or as a whole, their role in molecular recognition rather than discussing different terms in different literature would be more insightful.
5) In this review, the authors mentioned different forms of transient structure in different headings. The detailed information regarding their difference in terms of cut off distance within residue through space or authors only wanted to classify as they appeared in the different literature. A clear explanation would be helpful.
6) In the discussion, the reason stating the advantage of the NMR over SAXS and CD as almost all the cases, NMR is able to spot the transient structure whereas SAXS and CD are not. In my opinion, it will help beginner readers.
7) A schematic representation of the presence of “transient structures” along with a generalized mechanism for the protein-protein interaction and extended to drug discovery, if possible, would be helpful for the readers to follow the story of the review.
Minor corrections:
An intense care should be taken for the references as it showed some errors.
1) In reference 18 “ Hua, Q.-X.; Jia, W.-h.; Bullock, B.P.; Habener, J.F.; Weiss, M.A. Transcriptional ctivator_coactivator recognition: Nascent folding of a kinase-inducible transactivation domain predicts its structure on coactivator binding. Biochemistry, 1998, 37, 5858-5866. 451 DOI:10.1021/bi9800808 “…. I think that the highlighted part is “activator “instead of “ctivator”.
2) In line 141, Lee et al 2000 would be written as Lee et al (2000). In my opinion, it would be more convenient.
3) In reference 33, Zheng, Z.; Ma, D.; Yahr, T.L.; Chen, L. The transiently ordered regions in intrinsically disordered ExsE are correlated with structural elements involved in chaperone vinding. Biochem Biophys Res Commun. 2012, 417, 129-134. I think that it would be “binding”.
Author Response
In this paper named “Transient secondary structures in intrinsically disordered proteins”, authors bring a very interesting topic of the presence of transient secondary structures in IDPs and they addressed several works of literature in this regards. They also mentioned precisely the importance of these pre-populated transient secondary structures in the fulfillment of the
binding, recognition and ultimately the function. There are a few issues that I think to discuss more elaborately:
Thank you for helpful comments.
1) The authors put emphasis on pre-existing transient structure in free IDPs prior to target binding and they mentioned it as recognition antennae. I am wondering what it would be helpful for the reader to understand their point if they provide a well-defined mechanism for their proposal.
=> This point on the mechanistic view for IDP-target binding seems related to the comment #7. We have elaborated this point by providing a schematic diagram for binding. To date, two mechanisms of IDP-target binding, conformational selection and induced fit, have been proposed.
2) In this context, I would be interested to know about the author’s interpretation of the fuzzy complex of IDPs.
=> We have added a few sentences to describe the fuzzy complexes in lines 339-350.
3) In Line 122, authors mentioned the David Eliezer’s paper to address the term “preference for helical structure “in alpha-synuclein. I am curious to know whether “preference for helical structure “is equivalent to “having stable helical secondary structure” (as in Eliezer et al paper, far UV spectrum of alpha-synuclein in absence of SDS showed random coil-like structure) and
author’s explanation would be more useful for the general reader to understand this fine difference or similarities.
=> Judging the secondary structural content just by the appearance of CD spectra is not accurate. Careful analysis using secondary structure fitting programs will show that although CD spectra of IDPs appear structureless they do contain certain amount of secondary structures which are only transient, not stable. And these transient secondary structures can be accurately analyzed by NMR experiments. Hence, “preference for helical structure in the Eliezer’s paper is equivalent to having a transient helix, not “stable” helical secondary structure. Eliezer et al did state a “nascent or transient” helical structure, not a stable helix. We have added a sentence to clarify this point (lines 124-125).
4) Authors demonstrated the “transient secondary structure in IDPs “in various literature very efficiently. In this literature, different scientists interpret their (preformed structures) existence according to their problem. If I try to generalize the importance of their presence in protein-protein interaction, then the knowledge is still elusive. In my opinion, more literature, if any,
dealing with the acceleration of binding or enhance the specificity to the binding partner or as a whole, their role in molecular recognition rather than discussing different terms in different literature would be more insightful.
=> Please see #5.
5) In this review, the authors mentioned different forms of transient structure in different headings. The detailed information regarding their difference in terms of cut off distance within residue through space or authors only wanted to classify as they appeared in the different literature. A clear explanation would be helpful.
=> The purpose of this particular review article is indeed to present or list various expressions that have been used to denote transient structures in IDPs. The detailed definition of pre-structured motifs (PreSMos) and related descriptions have been presented in previous articles [reference 5 & 6 in the main text].
6) In the discussion, the reason stating the advantage of the NMR over SAXS and CD as almost all the cases, NMR is able to spot the transient structure whereas SAXS and CD are not. In my opinion, it will help beginner readers.
=> To help beginner readers who are non-NMR scientists we have input specific references that show this unique aspect of NMR in lines 311-314. We believe that interested readers can refer to these references.
7) A schematic representation of the presence of “transient structures” along with a generalized mechanism for the protein-protein interaction and extended to drug discovery, if possible, would be helpful for the readers to follow the story of the review.
=> We have prepared a generalized schematic diagram (Figure 1) explaining two target binding processes, conformational selection (CS) and induced fit (IF). In the figure legend we described how a PreSMo-derived peptide may become a peptide pharmaceutical as an inhibitor of target protein.
Round 2
Reviewer 2 Report
No Comment.